# Perception of a Haptic Stimulus Presented Under the Foot Under Workload

**DOI:** 10.3390/s20082421

**Published:** 2020-04-24

**Authors:** Landry Delphin Chapwouo Tchakoute, Bob-Antoine J. Menelas

**Affiliations:** Department of Mathematics and Computer Sciences, University of Quebec at Chicoutimi, Chicoutimi, QC G7H 2B1, Canada; bamenela@uqac.ca

**Keywords:** haptic stimulus, haptic communication, vibrotactile stimulus, haptic interfaces, haptics under workload, haptics via foot

## Abstract

It is clear that the haptic channel can be exploited as a communication medium for several tasks of everyday life. Here we investigated whether such communication can be altered in a cognitive load situation. We studied the perception of a vibrotactile stimulus presented under the foot when the attention is loaded by another task (cognitive load). The results demonstrated a significant influence of workload on the perception of the vibrotactile stimulus. Overall, we observed that the average score in the single-task (at rest) condition was greater than the overall mean score in the dual-task conditions (counting forwards, counting backwards, and walking). The walking task was the task that most influenced the perception of the vibrotactile stimulus presented under the foot.

## 1. Introduction

Humans constantly interact with their environment, which exposes them to conditions that can be distracting (public places, walking, metro, commercial spaces, etc.). In motion, for example, humans’ attention is mostly occupied by tasks such as walking while visualizing their space or walking while speaking to someone. These activities and/or tasks mostly involve haptic perception through the haptic channel [1]. Indeed, haptic perception is a process mediated by cutaneous and kinesthetic afferent subsystems [2]. However, we are now observing an increasing need for haptic interactions in everyday products to improve the user experience [3]. For instance, in designing, constraints of miniaturization and portability may considerably limit computation capabilities. This has an effect on the quality and the perception of the transmitted signal, especially in noisy environments [1]. Moreover, research in human–computer interaction (HCI) has shown the importance of choosing the haptic modality as a suitable channel for communication with humans [4,5,6]. We realized that in the area of health, the haptic channel could be used to orient people with low mobility [7]. It can also be used to reduce the risk of falling at home or in mobile situations by taking into account some aspects of the external environment, such as the types of soil [7,8]. In this context, in the past, we have demonstrated that it is possible to use the haptic channel to inform the user of a potential risk (low, medium, high, and very high) by using a vibrotactile stimulus sent to the plantar surface of the user’s foot [4]. Moreover, we have shown that the time taken to perceive a vibrotactile stimulus can vary on certain types of soil [9], and this variation was independent of the device positioning on the human body [10]. We have also found that the perception of a vibrotactile stimulus transmitted to the plantar surface of the foot can be influenced by auditory stimuli [9]. However, in everyday life, humans are constantly moving and spend most of their time engaged in dual-tasks situations (e.g., walking and talking on the phone), which can lead to focusing their attention on a specific task. One question thus arises about the perception of the transmitted information. Will the user be able to perceive this information; especially in the presence of external distractors? Thus, the main purpose of this study is to evaluate the haptic perception of information transmitted to the sole of the foot under workload (cognitive tasks and while walking).

This article is organized as follows. The introduction in Section 1 gives an overview of basic topics that are important for this study; this is followed by some related works in Section 2. Section 3 presents our research methodology and procedure, while Section 4 and Section 5 present the results and discussion, respectively. Section 6 presents the conclusion, with recommendations for future works.

## 2. Background

Attention is a complex cognitive function that is paramount in human behaviour. It allows for focusing on a main task while ignoring the other aspects, but it also involves cognitive efficiency, whether in perceiving, memorizing, or solving problems [11]. In everyday life, humans are often required to perform several tasks simultaneously, such as having a conversation while driving. Their attention can thus be shared between the main task and the secondary one. In this case, humans must concentrate on managing the many pieces of information because of the workload, and this requires more cognitive resources. A double-task situation may therefore require more resources, and this may cause a decline in performance in the main task [12].

When humans interact with computers, visual rendering is mainly exploited. Although this modality is suitable in many situations, there are scenarios in which the visual channel may be overloaded by the amount of presented data. To fully take advantage of the sensory capabilities of the human system, a benefit would certainly come from exploiting other sensory channels, such as the haptic and audio ones. Moreover, several studies support the use of haptic feedback as a means of communication [13,14] because, unlike visual feedback [15], it can improve human performance. Such observations explain why we are interested in using haptics as a communication under workload.

Oakley and Park have evaluated the recognition rate of tactons (structured vibrotactile messages) under workload [16]. The tactons were presented on a wearable device worn on the wrist while users performed three different distraction tasks. Two of the tasks involved completing work on a computer; the other was a mobile task requiring the participants to walk around. Tasks were chosen to represent common activities and explore different aspects of distraction. The results demonstrated the importance of the influence of cognitive disturbances. Tang et al. conducted a study to explore how haptic feedback could be used as another channel for transmitting information when the visual system is saturated [17]. They have shown that people can accurately perceive and process the haptic renderings of ordinal data under the cognitive workload. They evaluated three haptic models for rendering ordinal data with participants who were performing a taxing visual tracking task. The results demonstrated that information rendered by these models is perceptually available, even when users are visually busy. In the absence of other tasks, the participants detected and identified the change at 1.8 s and 2.5 s, respectively, but with the addition of visual and auditory distraction tasks, detection times increased significantly to 4.0 s [18]. In the same way, Chan et al. discuss the use of vibrotactile feedback to convey basic information when recipients are absorbed by a visual and/or auditory primary task [14].

Psychologists have studied various types of activity presented in daily living situations. For instance, Kaber and Zhang reviewed the use of virtual reality and haptic technologies for studying human performance in tasks involving the tactile sense [19]. Studies have revealed that tasks that are apparently dissimilar (e.g., talking and driving) can strongly interfere with each other when performed together [12] or in real-world challenges [20]. These tasks are mainly haptic tasks and mobility tasks [19]. Montero-Odasso et al. studied the walking task, a complex learned task that becomes automatic for most people from early childhood onwards and becomes a cognitive control of gait in older adults [21,22]. They hypothesized that walking while performing a secondary task (dual-task paradigm) is a way to assess the relationship between cognition and gait and is a dual-task paradigm for the daily living activities of elderly people [23]. In their procedure, they used counting forwards and counting backwards as the secondary task. At the same time, Timmermans et al. examined the effects of different walking environments on the cognitive-motor interference and task prioritization in dual-task walking in stroke patients [24]. They found that the walking environment strongly influenced the cognitive-motor interference and task prioritization during dual-task walking in stroke patients. All these studies evaluated various aspects of haptic communication in dual tasks and sometimes under workload. The major observation is that there is no clear study regarding the influence of a dual task and/or a cognitive load on haptic perception. In addition, these studies did not demonstrate how the workload influenced the perception of a haptic vibrotactile stimulus presented under the foot. The present study intended to fill this gap. We studied the perception of a vibrotactile stimulus presented under the foot when the attention is loaded by a main task (cognitive task or walking). To examine the validity of this hypothesis, we designed the following procedure.

## 3. Apparatus: Enactive Shoe

### 3.1. Wearable Device

The wearable device was designed based on some previous works [4,25,26]. We used an enactive shoe controlled by a smartphone that prevents falls related to a person’s immediate environment or an abnormal gait [8,27,28]. This device is a wearable device with a mini-computer-based processing system including a set of sensors and actuators distributed in strategic positions (Figure 1). It can be attached to the body, communicate wirelessly with smartphones (Figure 1f) via the Bluetooth and Wi-Fi protocols, and convey a vibrotactile stimulus to a specific location on the body during walking. The device has two separate enclosures, as suggested in [29]. Figure 1b is the processing system responsible for computing and conveying the signal, storing data, and waiting for the incoming participant input via the tactile screen. It is mainly composed of a Raspberry Pi-3-B mini-computer. Figure 1c is the haptic system responsible for transmitting the vibrotactile stimulus to a specific location of the body through haptuators. This is linked to the smartphone through a Bluetooth connection. The entire system is powered by a 9v lithium rechargeable battery of 600 mAh (Figure 1a). Each haptuator is mounted on removable straps that can be placed directly on a specific body location. The engine (haptuator) supply is a stable signal of 3.3 volts powered at a frequency of 100 Hz with an impedance of 10 ohms. This haptuator is not only capable of communicating, [4] but it is also capable of evaluating the impact of the auditory distraction [30] and the time taken to react to vibrotactile stimuli in the rest position [9]. Each haptuator measures 32 mm × 9 mm × 9 mm, with a frequency range between 90 and 1000 Hz [10].

### 3.2. Used Haptic Stimulus

One vibrotactile stimulus (Equation (1)) with a duration of one second was designed according to various studies [9,30,31,32] and was also used in our previous works [4,10]. The vibrotactile stimuli were elicited at frequency *w* (Equation (1)) and sent on Pacinian corpuscles mechanoreceptors field under the plantar surface of the foot.
*W* = a sin(2π ω t),(1)

## 4. Experiment

This experiment aimed to evaluate the influence of dual tasks on the perception of a vibrotactile stimulus conveyed randomly to the plantar surface of the foot.

### 4.1. Participants

Twenty-eight healthy students, aged from 20 to 35, from the University of Quebec at Chicoutimi (UQAC) participated in the study (Table 1). They were recruited by means of randomized sampling after a written electronic invitation to participate in a study related to the response time (RT) to a vibrotactile stimulus. All the participants attended the session voluntarily, and informed consent was obtained before the experimental sessions. Further, all the participants were novices to haptic technologies. Each participant filled out a short questionnaire on their health history and underwent touch inspection surrounding their foot sensitivity. The experiment and consent form were approved by the local ethics committee (certificate number: 602.434.01).

### 4.2. Experimental Setup

What follows is a description of the setup, including the protocol.

#### 4.2.1. Test Environment

The experimental phase took place in a calm space, specifically, in our laboratory at the university, equipped with chairs and a table for the preparation of the participants. The laboratory was equipped with the flooring surface conditions and a hygienic kit to clean the device after each session was finished. This environment remained constant during the experiments.

#### 4.2.2. Experiment conditions

To achieve the goal set in this study, we chose four tasks/activities that, based on preliminary studies [10,19,33], would highlight the influence of workload on haptic perception with the foot. We defined four conditions. The first condition was the at-rest condition, in which there was no distraction. The second was counting forwards to 100, the third was counting backwards from 100 to 0, and the fourth consisted of walking on the ground.

Our protocol is summarized in Table 2. We had two main sessions, which are described as follows:

### 4.3. Experimental Sessions: Control and Experimental

The experiments conducted to achieve the goal of this study involved two sessions: a control session and an experimental session. The control session concerned the evaluation of the vibrotactile stimulus perception in the normal condition (at rest, without any distraction), whereas the experimental session concerned the evaluation of vibrotactile stimulus perception under the influence of distractors. Each session (control and experimental) featured a familiarization phase followed by a test phase. For all conditions, the vibrotactile stimulus was conveyed when the participant had the left foot on the ground. The control session had an average duration of 25 min, whereas the experimental session lasted for around 45 min; there was a break of 5 min between the two sessions and between each phase. The space dimension used to walk was width × length (470 cm × 80 cm). The entire method (control and experimental) consisted of 336 measures (28 participants × 3 trials × 1 vibrotactile stimulus x 4 test phases). In all phases, a signal (vibrotactile stimulus) is sent. Each sending of a signal is spaced by at least five seconds and the user must have his foot on the ground. A delay of three seconds is granted for the user’s response. Otherwise, the answer is perceived to be a false positive.

### 4.4. Familiarization Phase

We had different configurations for the familiarization phase. It was performed at rest (sitting on a chair) for the baseline session and while walking for the control session. During this phase, we explained and demonstrated all aspects of the experiment to the participants. For the two conditions, the participants had ear protection and the haptic device attached to the left foot. We chose the left foot for technical reasons. Nothing in the literature suggests that there may be differences in lower extremity perception due to dominance. The participants had a smartphone on the hand outside of their field of vision. They also had to look at a black spot on the opposite wall. The participants were asked to press the smartphone screen whenever they perceived a vibrotactile stimulus. We recorded 28 participants × 1 vibrotactile stimulus × 3 trials (84 measures). Each good tacton recognition was computed as haptic perception recognition rate (score). Once each participant was trained (after achieving a recognition rate of 95%), the test phase began.

### 4.5. Test Phase

The test phase was performed at rest (sitting on a chair) for the baseline session. During this phase, the participants wore the haptic device and ear protection. For each condition phase, their vibrotactile stimulus perception was tested. When they perceived the vibrotactile stimulus, they pressed the smartphone screen. Thereafter, the identification was saved in the database. No results were shown to the participants during the test phase. All identifications were considered for measures. The test phase conditions were counterbalanced (randomized) between participants. When the test phase was finished, the participants were invited to fill in a form about the experience. The device was cleaned up after each test condition for the next participant.

## 5. Results

All the participants completed the experiment successfully, with 336 measures recorded (28 participants × 1 vibrotactile stimulus × 3 trials × 4 conditions). We took the mean of the three trials for each measure. Overall, we observed that the best mean perception score was achieved during the single-task condition (at rest; mean = 10.10 ± 0.73) and the worst was achieved in the presence of distractors (counting backwards; mean = 7 ± 2.21), as illustrated in Figure 2 (with mean value inside the bars). Between each condition, we observed some results on perception accuracy, as depicted in Figure 2. In the control condition (at rest), there was a mean score of 10.11 ± 0.72, which was the highest mean perception score among all conditions. For the dual-task conditions with cognitive tasks, there was an overall mean of 7 ± 2.17 in the counting forwards condition, compared to a mean of 6.39 ± 1.65 in the counting backwards condition. Finally, in the last dual-task condition with the motor task (walking), there was a mean of 5.75 ± 2.89, which was the smallest mean perception score among all conditions.

These results indicate that the perception accuracy was greater in the single-task condition than in the dual-task conditions, as illustrated in Figure 3 (with mean value inside the bars).

To detect any interaction effect between the conditions (single-task and dual-task), we performed an analysis of variance with repeated measures on the accuracy of the perception score. Indeed, as our goal was to examine the effect of dual tasks on the perception of a vibrotactile stimulus presented under the foot, we investigated whether distractions (dual task) had an impact on vibrotactile stimulus perception under the foot. Therefore, we made the following assumption:
**Hypothesis 1** **(H_1_).**Does a dual task affect vibrotactile stimulus perception?

We assumed that all the means would be equal if H_1_‘s null hypothesis (H_01_) were true and that at least one mean would be different from the others if H_1_’s alternative hypothesis (H_a1_) were true. Our significance level was (alpha) = 0.05. The dependent variable was the score, and our independent variables were the conditions. All the tests were performed using Minitab version 17, and the visualizations were created using Power BI Desktop, January 2019 release. The analysis of variance (ANOVA) evaluation was performed with post-hoc Tukey HSD (honest significant difference) tests. Our input data independent group factors were at rest (single task), counting forwards, counting backwards, and walking (dual task). Pairwise comparisons, which identified significant differences between conditions, were used in all analyses. Statistical significance was set at the 95% confidence level (*p* < 0.05). The sample observation was *n* = 28. The data satisfied the conditions to justify the use of ANOVA. Analysis of the distribution of the data suggested that they were normally distributed.

A one-way ANOVA with repeated measures was conducted to determine whether the perception of the vibrotactile stimulus under the foot was different for the participants in the dual-task condition. Overall, there was a statistically significant difference between the conditions, as determined by the result of the one-way ANOVA: F_(3,108)_ = 24.71, F_critical_ = 2.70, *p* < 0.05. This result suggests that one or more of the conditions (dual task) influenced the perception of the vibrotactile stimulus. According, to Cohen’s guidelines [34], a small effect size is 0.01, a medium effect size is 0.059, and a large effect size is 0.138. Therefore, the effect size we obtained, (η^2^) = 0.407, was a very large effect size. It also indicates that 41% of the change in the perception accuracy (score) can be accounted for by the conditions (dual task), as depicted in Figure 4 (the circles in the figure are mean values). With a Cohen’s d [34] of 0.9, 82% of the participants in the dual-task condition, counting forwards, counting backwards, and walking, will be above the mean of the participants at rest, without a dual task. Further, 65% of the two groups will overlap, and there is a 74% chance that a person chosen at random from the dual-task group will have a higher perception score (accuracy) than a person selected at random from the group without the dual task (probability of superiority). Moreover, to have one outcome that is more favorable in the dual-task group than in the single-task group (at rest), we need to treat 3.1 people. This means that if 100 people perceived the vibrotactile stimulus in the dual-task condition, 32.3 more people would have a more favorable score than if they had perceived the stimulus at rest.

To identify the conditions that most influenced the perception of the vibrotactile stimulus, we performed the Tukey simultaneous test for differences of means (Table 3).

We observed a significant difference between the single-task condition and the three groups of dual-task conditions (Table 3). This means that the dual tasks (cognitive and motor) used in this experiment decreased the perception accuracy of the vibrotactile stimulus presented under the foot. However, as illustrated in Table 3, the difference between the dual tasks was not significant.

We performed an analysis on false positive to see how user’s responses referred to the stimulus delivered. We performed an ANOVA analysis on false positive response between conditions. We observed a non-significant difference between all conditions. This means that participant’s answers really referred to the delivered vibrotactile stimulus.

We also analyzed the normality of the sample checked. We used the Anderson–Darling test. Results diagrams and plots are reported in the Figure 5.

## 6. Discussions

The objective of this study was to evaluate the influence of dual tasks on the perception of a vibrotactile stimulus presented under the foot. The data were analyzed in terms of the overall mean score of perception within each condition, single task (at rest) and dual tasks (counting forwards, counting backwards, and walking). The results obtained in this study indicate that being in a dual-task condition is detrimental to the perception of stimuli presented under the foot.

### 6.1. Influence of Walking on the Perception of a Vibrotactile Stimulus

During our study, we mainly used two tasks: pure cognitive (to count) and a cognitive-motor task (to walk). The results revealed a significant difference in response time to the perception stimulus in the walking task. According to Table 3, the walking task represented the worst mean score obtained by the participants; this indicates that walking can affect perception when a user perceives a vibrotactile stimulus under the foot. This result can be explained by the fact that walking is a complex motor act, a multisegmental task and a semi-automatic control of rhythmic movements by the central nervous system [10]. This implies that sensory information (haptic and other) coming from the lower limbs reaches the sensory cortex during the walking motor task using a neurological network. This sensory information can facilitate the perception when it is a repetitive task (motor coordination) or bias vibrotactile stimulus perception when it is not a repetitive task. Moreover, the interpretation of psychophysiological responses becomes doubly difficult when a person is engaged in a multi-task environment [12]. Indeed, it has been shown that in daily life, tasks that are apparently dissimilar (e.g., talking and driving) can strongly interfere with each other when done together, both in laboratory conditions [35] and in real-world challenges [20]. One of the most useful methods for studying such situations is the dual-task paradigm. This paradigm involves performing two tasks concurrently, resulting in impaired behavioural performance on one or both tasks. As it was in our experiment, we found that the walking represented the task that most influenced the perception of a vibrotactile stimulus presented under the foot. This finding is in line with some other works, which have reported that vibrotactile stimuli delivery to the foot was influenced in communicating information in some conditions [4,9,10,28,29]. While most of the participants reported the system to be comfortable, they all reported that the walking condition was much more difficult than the others.

### 6.2. Influence of Cognitive Tasks on Vibrotactile Stimulus Perception

As pointed out previously, there was a significant difference between vibrotactile stimulus perception at rest (single task) and vibrotactile stimulus perception when the participants were counting forwards and/or counting backwards (dual task). However, the results of the difference levels (Table 3) for counting forwards and backwards had an adjusted *p*-value of 0.272, which is not significant. This is not surprising, because the dual-task counting forwards or backwards is related to the same cortex in memory as cognitive tasks [36]. Thus, on average, there was no difference of perception when the participants were engaged in a cognitive dual task because they always remained with the same workload. In other words, the dual tasks: motor (walking) or cognitive task (counting forwards and counting backwards) can be associated to the same workload when delivering a stimulus on the foot. This is in line with some other works [12,37], which have found that perception can be discriminate between normal performance and mental overload. Like it was in our study, the two tasks can be expected to at least partially share the same mental resources, since they are both arithmetic tasks [36]. Our results tend to be in line with the work of Oakley and Park, who found that distraction (double or multitasking) masks the perception of vibrotactile signals and influences the main task [16]. In summary, our results point out the influence of cognitive tasks; one could state that cognitive distractors should be taken into account when designing and conveying information with the foot, as mentioned in [16]. All this reinforces the idea that the haptic channel can be used as a communication channel [38,39].

### 6.3. Implication of the Results from This Study

The results of this study show a poor performance in vibrotactile messages perception when walking, unlike cognitive tasks. These results indicate that it is not appropriate to communicate a risk of falling throughout a haptic message presented under the foot as suggested in [1,5]. Indeed, when walking there is friction exerted on the receptors of the surface of the foot. These frictions can cause perceptual conflicts, especially when the sole has different textures [8,9,10]. As a result, the transmission of information by the haptic modality in such conditions will be poorly perceived because of the dual-task situation. As a result, a future improvement could underline a new method who could take into account these results. One approach would be to choose another position that does not suffer from the perceptual conflict and would have a better haptic perception rate. To this, the literature offers connected jackets to communicate non-visual information [40,41,42,43] or the wrist [16]. This new method will have a wider range of applications including spatial orientation and guidance, attention management, haptic guidance, assistance, and better learnability [44,45].

## 7. Conclusions and Future Works

Following the design and development of a shoe capable of preventing accidental falls linked to balance problems, we studied the use of vibrotactile returns to communicate a risk of falling. The objective of this study was to investigate the influence of workload on the perception of a vibrotactile stimulus presented on under the foot. The apparatus used was a wearable device capable of conveying a vibrotactile stimulus. The experimentation protocol included four conditions: one single task and three dual tasks. The measures were performed in the controlled environment of our laboratory. Overall, the results revealed a significant influence of dual tasks on the perception of a vibrotactile stimulus. Specifically, we observed that the dual task involving walking influenced the perception of the vibrotactile stimulus far more than the cognitive task (counting forwards or backwards). These findings suggest that dual tasks could bias perception and should be taken into account when conveying vibrotactile stimuli under the foot. For this, as an alternative we intend to investigate the use of clothing for the communication of haptic information. This avenue is privileged as it appears that this new method will have a wider range of applications including spatial orientation and guidance, attention management.

## Figures and Tables

**Figure 1 sensors-20-02421-f001:**
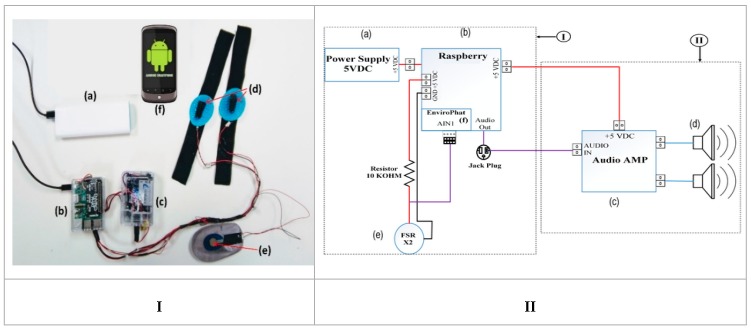
The wearable device worn on the left foot with a strap to hold the haptuator. (**I**) The device component used for the experiment. The haptuator is located under the arch of the second toe fixed by the black strap. (**II**) The electronic diagram of the device showing how the components are joined together in order to deliver the vibrotactile stimulus.

**Figure 2 sensors-20-02421-f002:**
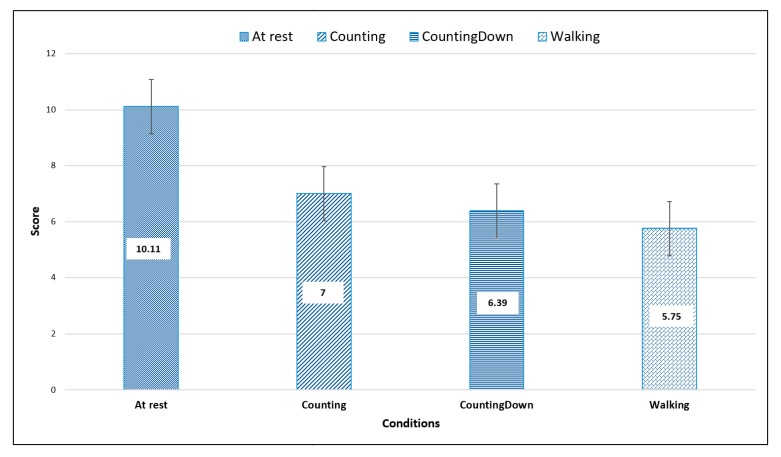
Score mean perception of each condition (with mean value inside the bars).

**Figure 3 sensors-20-02421-f003:**
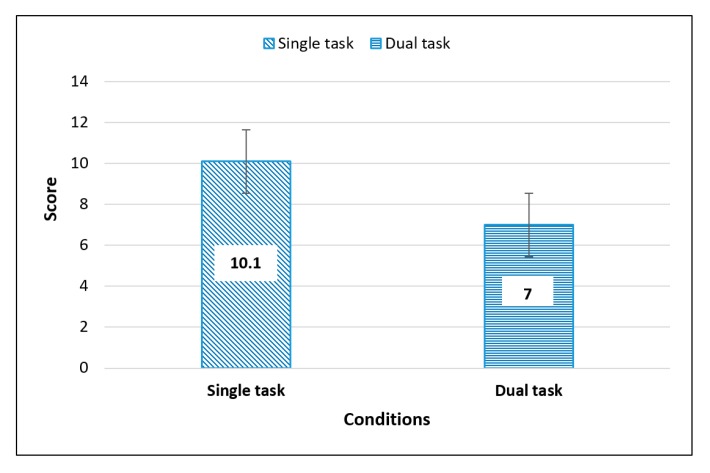
Overall score mean perception of single task compared to dual task (with mean value inside the bars).

**Figure 4 sensors-20-02421-f004:**
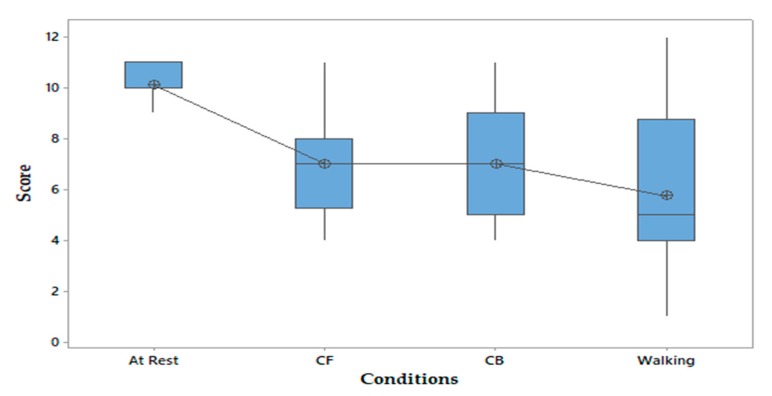
Box plot of score perception between conditions: at rest; counting forwards (CF); counting backwards (CB); and walking.

**Figure 5 sensors-20-02421-f005:**
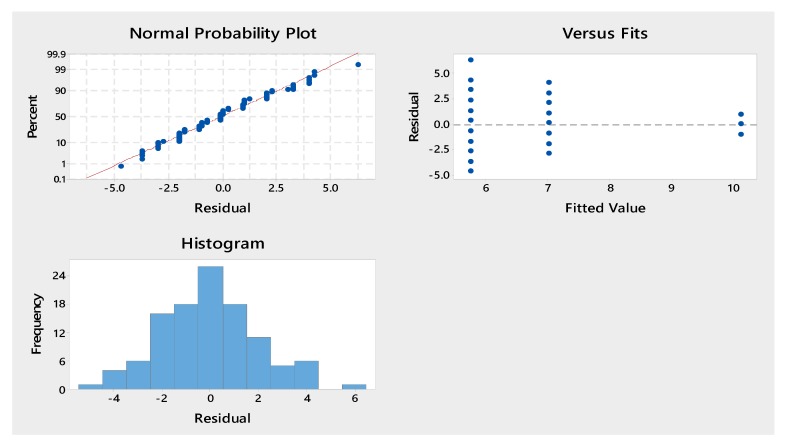
Normality test results: residual plots of conditions at rest, counting forward (CF), counting backward (CB), and walking.

**Table 1 sensors-20-02421-t001:** Participant’s characteristics.

Participants	Value
Age (Y)	26.45 ± 4.45 *
Height (cm)	162.95 ± 29.42 *
Weight (kg)	78.98± 33.26 *
Gender	Men (*n* = 14)|Women (*n* = 14)

* Values represented as mean ± standard deviation (SD).

**Table 2 sensors-20-02421-t002:** Experiment protocol summary.

Sessions	Conditions	Phase	Distractions	Positioning
Control	1: At rest	Familiarization phase	None	Static: At rest
		Test phase	None	Static: At rest
Experi-mental	2: Counting forwards	Familiarization phase	Counting Forwards	At rest
		Test phase	Counting Forwards	At rest
	3: Counting backwards	Familiarization phase	Counting Backwards	At rest
		Test phase	Counting Backwards	At rest
	4: Walking	Familiarization phase	Walking	Moving
		Test phase	Walking	Moving

**Table 3 sensors-20-02421-t003:** Tukey simultaneous tests for differences of means.

Difference of Levels	Difference of Means	SE of Difference	95% CI	T-Value	Adjusted *p*-Value
CF–At rest	−3.107	0.550	(−4.541; −1.673)	−5.65	0.000
CB–At rest	−3.714	0.550	(−5.148; −2.280)	−6.76	0.000
Walking–At rest	−4.357	0.550	(−5.791; −2.923)	−7.93	0.000
CB–CF	−0.607	0.550	(−2.041; 0.827)	−1.10	0.687
Walking–CF	−1.250	0.550	(−2.684; 0.184)	−2.27	0.11
Walking–CB	−0.643	0.550	(−2.077; 0.791)	−1.17	0.647

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
