# Peer review of "Perception of a Haptic Stimulus Presented Under the Foot Under Workload"

_sensors, 2020, doi:10.3390/s20082421_

Round 1

Reviewer 1 Report

This manuscript presented a series of studies to evaluate the capability of human subjects to discriminate vibrotactile stimulations provided under the foot during the development of another task (i.e. dual tasks experiment). The concurrent tasks could be cognitive (forwards and backwards counting) or motor (walking) ones. The performances obtained during the dual tasks were compared with those ones found in the vibrotactile stimuli discrimination task at the rest (single task). Best performances were obtained for the single task and worst performance for motor dual task. A significant difference between the two cognitive dual tasks was not found. Such study can be useful for the development of system of alarm and/or navigation studying the cognitive load to employ to recognize the occurred stimulation. The manuscript is of interest; however, some points in the method should be better explained and the discussion should take into account some limitation of the experimental method.

The experiments were performed with the same order of the conditions for each participant, this could affect the findings: in particular, the results with dual task could be worse if these trials were performed first because there is not effect of training, or also the fatigue could affect the performance. This should be mentioned in the discussion. Correctly, during the motor dual task, the stimulation was released when the foot was in contact with the ground, to allow an optimal contact with the stimulation device; however, during the walking activity, also when the foot is on the ground, this is on movement and this fact could affect the discrimination of the vibrotactile stimulation. This should be mentioned on the discussion. At the beginning of the manuscript the authors talked about just dual tasks; however it is evident that cognitive and motor tasks are different (in motor task there is cognitive load and also movement of the part that receives the stimulation). The authors should split from the beginning of the manuscript the dual tasks in cognitive ones and motor one. It is not clear to me whether the authors evaluated false positive, how are they sure that the responses referred to the stimulus delivered? Did they take the time of response of the participants? Was the normality of the analyzed samples checked? Which tests were employed? The authors talked about perception score and perception accuracy, are they the same outcome? In general, it should be explained what they correspond. It is not clear the frequency of stimulation of the actuator (vibrotactile device), 100Hz seems to be the frequency of supply of the actuator, not the frequency of stimulation. Equation 1 “121” should be “ω”. Table 3 and Figure 3 provided the same information, I suggest to eliminate the table, it is repetitive. In Figure 3 and 4 it is not clear what is the number texted on the bars, this should be specified. If this is the mean value, in figure 4 the shown values are not correct. In figure 4 missing the error bars. In figure 5 it is not clear what are the circles, are they the mean values? this should be specified. The sentences “we found that the walking represented the task that most influenced the perception of a vibrotactile stimulus presented under the foot. This finding is in line with some other works, which have reported that compared to a visual-only condition, tacton delivery to the foot was successful in communicating information or in reducing stress and errors.” were not clear and seem to be conflicting between them. Also the sentences “there was no difference of perception when the participants were engaged in a cognitive dual task because they always remained with the same workload. In other words, the lack of difference between the counting forwards and backwards conditions may have been caused by mental overload.” .” were not clear and seem to be conflicting between them. Caption Figure 1: the sentence “how the component are join together” should be “how the components are joint together”. Title “32. Used Haptic Stimulus” should be “3.2. Used Haptic Stimulus”

Author Response

MDPI 2020 Revision

Perception of a Haptic Stimulus Presented Under the Foot Under Workload

Reviewer 1:

I would not like to sign my review report

English language and style: Exten(x) Moderate English changes required

Does the introduction provide sufficient background and include all relevant references? Yes

Is the research design appropriate? Can be improved

Are the methods adequately described? Must be improved

Are the results clearly presented? Must be improved

Are the conclusions supported by the results? Must be improved

Comments and Suggestions for Authors

Resume: This manuscript presented a series of studies to evaluate the capability of human subjects to discriminate vibrotactile stimulations provided under the foot during the development of another task (i.e. dual tasks experiment). The concurrent tasks could be cognitive (forwards and backwards counting) or motor (walking) ones. The performances obtained during the dual tasks were compared with those ones found in the vibrotactile stimuli discrimination task at the rest (single task). Best performances were obtained for the single task and worst performance for motor dual task. A significant difference between the two cognitive dual tasks was not found. Such study can be useful for the development of system of alarm and/or navigation studying the cognitive load to employ to recognize the occurred stimulation.

The manuscript is of interest; however, some points in the method should be better explained and the discussion should take into account some limitation of the experimental method.

Q1: The experiments were performed with the same order of the conditions for each participant, this could affect the findings: in particular, the results with dual task could be worse if these trials were performed first because there is not effect of training, or also the fatigue could affect the performance. This should be mentioned in the discussion.

A1: In fact, the experiments were performed randomized. We reported this information at Section 4.5: “The test phase conditions were counterbalanced (randomized) between participants”. For a better clarity purpose, we added this information in protocol and discussion sections.

Q2: Correctly, during the motor dual task, the stimulation was released when the foot was in contact with the ground, to allow an optimal contact with the stimulation device; however, during the walking activity, also when the foot is on the ground, this is on movement and this fact could affect the discrimination of the vibrotactile stimulation. This should be mentioned on the discussion.

A2: We agree, this information has been added into the discussion

Q3: At the beginning of the manuscript the authors talked about just dual tasks; however it is evident that cognitive and motor tasks are different (in motor task there is cognitive load and the movement from the part that receives the stimulation). The authors should split from the beginning of the manuscript the dual tasks in cognitive ones and motor one.

A3:  For clarity, we speak rather about the workload. Therefore, we have cognitive distractions and walking.

Q4: It is not clear to me whether the authors evaluated false positive, how are they sure that the responses referred to the stimulus delivered? Did they take the time of response of the participants? Was the normality of the analyzed samples checked? Which tests were employed?

A4:

  1. We use this protocol. Each sending of a signal is spaced by at least five seconds and the user must have his foot on the ground. A delay of three seconds is granted for the user's response. Otherwise, the answer is perceived to be a false positive. The statistical analysis made on these false positive data were not significant.
  2. The test of the normality was analyzed on the sample. We used the Anderson-Darling Test. Here is the result of the test analysis:

Figure 1  Normality Test

Q5: The authors talked about perception score and perception accuracy, are they the same outcome? In general, it should be explained what they correspond.

A5: Both relate to the same outcome. We have standardized our words for clarification

Q6: It is not clear the frequency of stimulation of the actuator (vibrotactile device), 100Hz seems to be the frequency of supply of the actuator, not the frequency of stimulation.

A6: Yes, you are right. It is not the frequency of stimulation, but the one of the supply. That’s why, in section 3.1 we mentioned: The engine (haptuator) supply is a stable signal of 3.3 volts powered at a frequency of 1000 Hz with an impedance of 10 ohms.

Q7: Equation 1 “121” should be “ω”.

A7: We changed this

Q8: Table 3 and Figure 3 provided the same information, I suggest to eliminate the table, it is repetitive.

A8:  Thank you. I delete the table 3 and changed the label of table 4 to table 3 also the caption within the manuscript

Q9: In Figure 3 and 4 it is not clear what is the number texted on the bars, this should be specified.

A9: The numbers on the bars are mean value. We added the information in the manuscript at the first reference of the figure.

Q10: If this is the mean value, in figure 4 the shown values are not correct. In figure 4 missing the error bars.

A10: We changed the figure 4 and we added the error bar.

Q10: In figure 5 it is not clear what are the circles, are they the mean values? this should be specified.

A10: We added the information in the manuscript at the first reference of the figure.

Q11: The sentences “we found that the walking represented the task that most influenced the perception of a vibrotactile stimulus presented under the foot. This finding is in line with some other works, which have reported that compared to a visual-only condition, tacton delivery to the foot was successful in communicating information or in reducing stress and errors.” were not clear and seem to be conflicting between them.

A11: Thank you for this observation. We changed the sentences with this one: “we found that the walking represented the task that most influenced the perception of a vibrotactile stimulus presented under the foot. This finding is in line with some other works, which have reported that vibrotactile stimuli delivery to the foot was influenced in communicating information in some conditions [4,9,10,26,29]”.

Q12: Also the sentences “there was no difference of perception when the participants were engaged in a cognitive dual task because they always remained with the same workload. In other words, the lack of difference between the counting forwards and backwards conditions may have been caused by mental overload.” .” were not clear and seem to be conflicting between them.

A12: We changed the sentence and we added this one: In other words, the dual tasks (counting forwards and counting backwards) either motor or cognitive task can be associated to the same workload when delivering a stimulus on the foot.

Q13: Caption Figure 1: the sentence “how the component are join together” should be “how the components are joint together”.

A13: We changed this, and we add “s” to the word component

Q14: Title “32. Used Haptic Stimulus” should be “3.2. Used Haptic Stimulus”

A14: We changed this by adding the “.” between 3 and 2

Reviewer 2 Report

This paper study the effect of different types of tasks towards the perception of haptioc feedback. While the authors gave interesting motivation of the study, the following points shall be revised to enhance the quality of the paper.

Major:
1. Clear definition of perception score is not given.
2. The correlation between accuracy of primary task and perception score
shall be investigated to rule out the possibility of subject not concentrate
enough in primary task.
3. It would also be interesting to analyze the outcome of familiarization phase and
how it helps in test phrase.
4. Discussion on implication of the result from the study should be provided.

Minor:

1. In introduction, the tense in English usage is not consistent across the section.
2. Table 2 should be more dense to save the space.

Author Response

MDPI 2020 Revision

Perception of a Haptic Stimulus Presented Under the Foot Under Workload

Reviewer 2:  

I would not like to sign my review report

English language and style: Moderate English changes required

Does the introduction provide sufficient background and include all relevant references? Yes

Is the research design appropriate? Can be improved

Are the methods adequately described? Must be improved

Are the results clearly presented? Can be improved

Are the conclusions supported by the results? Must be improved

Comments and Suggestions for Authors

Resume: This paper study the effect of different types of tasks towards the perception of haptic feedback. While the authors gave interesting motivation of the study, the following points shall be revised to enhance the quality of the paper.

Major:

Q1: Clear definition of perception score is not given.

A1: Thank you for this observation. The perception score refers to the the tacton recognition rate of each participant. To clarify this, we add the sentence: “Each good tacton recognition is computed as haptic perception recognition rate (score).” in section 4.4. In fact, this expression has been already used in some studies [1,2]

References:

  1. Menelas, B. A. J., & Otis, M. J. D. (2012, October). Design of a serious game for learning vibrotactile messages. In 2012 IEEE International Workshop on Haptic Audio Visual Environments and Games (HAVE 2012) Proceedings (pp. 124-129). IEEE.
  2. Tchakouté, L. D. C., Gagnon, D., & Ménélas, B. A. J. (2018). Use of tactons to communicate a risk level through an enactive shoe. Journal on Multimodal User Interfaces, 12(1), 41-53.

Q2: The correlation between accuracy of primary task and perception score shall be investigated to rule out the possibility of subject not concentrate enough in primary task.

A2:   Thank you for this recommendation. The purpose of this study was to analyze the influence of some external factors while perceiving a haptic signal. Here we have chosen three human daily activities involving cognitive load (walking, counting, and counting down). The context here is that, when a user is in daily activity, he is exposed to some factors or conditions that could influence the haptic perception. Regarding, the purpose of this study, we wanted to answer the question, how everyday activities may influence the perception of a haptic stimulus?  That’s why we reframe within the manuscript “dual task” as “cognitive load task” which can include tasks (conditions) like walking, counting, and counting down. 

Q3: It would also be interesting to analyze the outcome of familiarization phase and how it helps in test phrase.

A3: The outcome of the familiarization has been already evaluated in our previous paper [1] and the result was that the repetition significantly improves the recognition rate of tactons but does not speed up the completion time. The present study is like a continuation of the study [1].

In [1], we have just evaluated the communication of stimuli whereas in the present study we have shown that an external task can influence the perception of one stimulus.

References:

[1] Tchakouté, Landry Delphin Chapwouo, David Gagnon, and Bob-Antoine Jerry Ménélas. "Use of tactons to communicate a risk level through an enactive shoe." Journal on Multimodal User Interfaces 12.1 (2018): 41-53.

Q4: Discussion on implication of the result from the study should be provided.

A4:  We added a new section 5.5 with the following sentence:

In everyday life, humans perform walking in various ways.). Walking is a motor and cognitive task used constantly. Our results in this study show a poor performance in vibrotactile messages perception when walking, unlike cognitive tasks. These results show that it is not appropriate to communicate a risk a falling throughout a haptic message presented under the foot as suggested in [1, 5]. Indeed, when walking there are frictions exerted on the receptors under the foot. These frictions can cause perceptual conflicts, especially when the sole has different textures [1]. As a result, the transmission of information by the haptic modality in such conditions will be poorly perceive because of the dual-task situation.

Thus, a future improvement could underline a new method who could take into account these results. One approach would be to choose another position that does not suffer from the perceptual conflict and would have a better haptic perception rate. To this, the literature offers connected jackets to communicate non-visual information [1,2,3,4]. This new method will have a wider range of applications including spatial orientation and guidance, attention management, haptic guidance, assistance, and sensory substitution.

References:

  1. Otis, M. J. D., Ayena, J. C., Tremblay, L. E., Fortin, P. E., and Menelas, B.-A. J. (2016). Use of an enactive insole for reducing the risk of falling on different types of soil using vibrotactile cueing for the elderly. PLOS ONE, 11(9):1–26.
  2. A. Jones, M. Nakamura and B. Lockyer, "Development of a tactile vest," 12th International Symposium on Haptic Interfaces for Virtual Environment and Teleoperator Systems, 2004. HAPTICS '04. Proceedings., Chicago, IL, USA, 2004, pp. 82-89.
  3. Erp, J. B. V., Veen, H. A. V., Jansen, C., & Dobbins, T. (2005). Waypoint navigation with a vibrotactile waist belt. ACM Transactions on Applied Perception (TAP), 2(2), 106-117.
  4. Karuei, I., MacLean, K. E., Foley-Fisher, Z., MacKenzie, R., Koch, S., & El-Zohairy, M. (2011, May). Detecting vibrations across the body in mobile contexts. In Proceedings of the SIGCHI conference on Human factors in computing systems (pp. 3267-3276).
  5. Tchakouté, Landry Delphin Chapwouo, David Gagnon, and Bob-Antoine Jerry Ménélas. "Use of tactons to communicate a risk level through an enactive shoe." Journal on Multimodal User Interfaces 12.1 (2018): 41-53.

Minor:

Q1. In introduction, the tense in English usage is not consistent across the section.

A1: Thank you for this observation. Changes have been made.

Q2. Table 2 should be more dense to save the space.

A2: Thank you for this observation. Changes have been made.

Round 2

Reviewer 1 Report

The manuscript after the revision resulted to be improved; however I have other comments and formal points

Comments:

In the response A4 to my comment the authors reported:

  1. Each sending of a signal is spaced by at least five seconds and the user must have his foot on the ground. A delay of three seconds is granted for the user's response. Otherwise, the answer is perceived to be a false positive. The statistical analysis made on these false positive data were not significant.
  2. The test of the normality was analyzed on the sample. We used the Anderson-Darling Test.

The authors should introduce such information also on the main test

The frequency of stimulation of haptuator is missing

There is something of odd in this sentence “In other words, the dual tasks (counting forwards and counting backwards) either motor or cognitive task can be associated to the same workload when delivering a stimulus on the foot.” Why did the authors introduce motor task when they talked just about the cognitive tasks in the paragraph? They should introduce before the sentence that also between cognitive and walking task there is not significant difference in performance, and in the following sentence would write “In other words, the dual tasks: motor (walking) or cognitive task (counting forwards and counting backwards) can be associated to the same workload when delivering a stimulus on the foot.”

Formal points

  • In caption of Figure1 also the final “ed” in “join” is missing
  • Many parts of text between parentheses added in this version of the work started with capital letter please correct it

Author Response

Comments:

Q1: In the response A4 to my comment the authors reported:

Each sending of a signal is spaced by at least five seconds and the user must have his foot on the ground. A delay of three seconds is granted for the user's response. Otherwise, the answer is perceived to be a false positive. The statistical analysis made on these false positive data were not significant. The test of the normality was analyzed on the sample. We used the Anderson-Darling Test. The authors should introduce such information also on the main test

A1: We introduce this information in the manuscript. The first part: “In all phases, a signal (vibrotactile message) is sent. Each sending of a signal is spaced by at least five seconds and the user must have his foot on the ground. A delay of three seconds is granted for the user's response. Otherwise, the answer is perceived to be a false positive.” has been added into the section 4.3. The second part : “”  has been added into the section 5. With the figure numbered Figure 1

Q2: The frequency of stimulation of haptuator is missing

A2: The following sentence has been added at the end of section 3.1: “Each haptuator measures 32 mm × 9 mm × 9 mm, with a frequency range between 90 to 1000 Hz [10].”

Q3: There is something of odd in this sentence “In other words, the dual tasks (counting forwards and counting backwards) either motor or cognitive task can be associated to the same workload when delivering a stimulus on the foot.” Why did the authors introduce motor task when they talked just about the cognitive tasks in the paragraph? They should introduce before the sentence that also between cognitive and walking task there is not significant difference in performance, and in the following sentence would write “In other words, the dual tasks: motor (walking) or cognitive task (counting forwards and counting backwards) can be associated to the same workload when delivering a stimulus on the foot.”

A3: Thank you very much for the suggestion. We follow your point and we changed the sentence.

Formal points

Q4: In caption of Figure1 also the final “ed” in “join” is missing

A4: Thank you very much for the suggestion, we changed the sentence.

Q5: Many parts of text between parentheses added in this version of the work started with capital letter please correct it.

A5: Thank you very much for the suggestion, we changed the sentence.
